# ACTIONS SPEAK LOUDER THAN STATES:
# GOING BEYOND BAYESIAN INFERENCE IN
# IN-CONTEXT REINFORCEMENT LEARNING

## ABSTRACT

This paper explores the emergence of in-context learning (ICL) in reinforcement learning (RL) environments, focusing on how transformers can surpass Bayesian inference limitations. We investigate the critical role of task diversity in enabling transformers to develop advanced learning algorithms for RL. To overcome the limitations of existing RL environments in providing sufficient task variety, we introduce a novel benchmark based on the Omniglot dataset, offering unprecedented task diversity. Through extensive experimentation, we demonstrate that increasing task diversity leads to significant improvements in transformers' ability to generalize to unseen tasks. We examine the effects of model capacity, regularization techniques, and action representation on ICL performance. Our findings reveal that larger model capacities and specific augmentation strategies contribute to enhanced ICL capabilities. Notably, we observe a clear transition from Bayesian inference-like behavior to more advanced learning paradigms as task diversity increases. This research provides crucial insights into the factors driving generalizable in-context reinforcement learning in transformers and underscores the importance of designing RL environments with qualitatively diverse tasks to unlock the full potential of ICL in RL scenarios.

## 1 INTRODUCTION

Transformers (Vaswani et al., 2017), with their widespread applications across numerous fields and substantial evidence of effectiveness, have become a cornerstone in modern machine learning. These models, particularly when pretrained on extensive text corpora, have demonstrated remarkable proficiency in transferring knowledge to related downstream tasks through finetuning on smaller, task-specific datasets (Devlin et al., 2018; Howard & Ruder, 2018; Radford et al., 2019). A striking capability emerges when these pretrained transformers are exposed to training datasets characterized by high task diversity: they adapt to new tasks directly through the examples in their context, without the need for further training (Brown et al., 2020). This phenomenon, termed in-context learning (ICL), contrasts starkly with other settings such as supervised learning (SL) and reinforcement learning (RL), where acquiring large, diverse datasets poses a significant challenge (Levine et al., 2020).

The exploration of in-context learning within supervised learning, particularly in regression and classification, has been extensively documented (Kirsch et al., 2022; Garg et al., 2022; Zhang et al., 2023; Wies et al., 2023; Von Oswald et al., 2023; Li et al., 2023; Bai et al., 2023). It has been proposed that transformers might implicitly learn various learning algorithms, such as ridge regression, gradient descent, and Bayesian inference, influenced by factors like task diversity and transformer architecture (Xie et al., 2021; Akyürek et al., 2022; Von Oswald et al., 2023). Moreover, training regimes that deviate from Bayesian principles have been observed with increased task diversity (Raventós et al., 2023) or the distribution of training data (Chan et al., 2022).

In the realm of reinforcement learning, in-context learning has garnered attention, particularly where models are prompted with trajectories or timestep transitions in the hope of learning from demonstrations. Signs of ICL in reinforcement learning has been shown to emerge through direct training (Melo, 2022), prompt tuning (Hu et al., 2023), supervised training (Lee et al., 2023), and leveraging

pretrained large language models (Reid et al., 2022; Li et al., 2022). Team et al. (2023) studies task diversity in a closed-source domain where they do not share their trained models nor training code. Furthermore, its online training of its agents renders its results unrelated for many domains that can mostly focus on offline training (e.g., robotics). Lu et al. (2024) is another work that focuses on online training. However, they also note that they achieve limited generalization to unseen control tasks. Kirsch et al. (2023) train their agent as more of a learning-to-RL style on offline trajectories. Their choice of simple augmentation to increase task diversity again results in limited performance on unseen tasks. Raparthy et al. (2023) shows transformers trained on a sequence of expert trajectories can have limited generalization to unseen game levels. Furthermore, this work does not show the scaling with massive task diversity. Since they only have at most 12-16 tasks, they cannot reliably show the scaling as we can show in our experiments for more than 16k tasks. Despite these efforts, a comprehensive understanding of the generalization capabilities of these models, particularly the factors contributing to such capabilities, remains elusive. While the Bayesian inference regime has been achieved in some instances (Laskin et al., 2022; Lee et al., 2023; Lin et al., 2023), the generalization to unseen tasks continues to be a significant hurdle (Li et al., 2022; Liu et al., 2023). Therefore, as far as we are aware, our work is the first that can show this transition with respect to the number of tasks used in the pretraining.

In RL, agents learn by interacting with an environment, where they must make a series of decisions that affect future states and rewards (Sutton & Barto, 2018). This dynamic nature introduces complexities such as temporal credit assignment, where agents must learn which actions lead to rewards over time (Mnih et al., 2015), and the exploration-exploitation dilemma (Auer et al., 2002; Bellemare et al., 2016), which requires balancing the discovery of new strategies against optimizing known ones. The context in RL, therefore, involves understanding not just the immediate outcomes of actions, but also their long-term effects and dependencies (De Asis et al., 2018). This makes in-context learning in RL significantly more challenging, as models must infer not only the immediate consequences of actions but also their impact on future states and rewards. Meta-RL shows that learning new but similar environments can still be quite challenging even with the ability to finetune the model. Therefore, if the environments are separate enough, we cannot hope to learn new policies for those environments quickly in the in-context RL setting as well. Therefore, we utilize a common idea in the NLP domain for LLMs, prompting with demonstrations. In ICRL, the demonstrations are expert trajectories and the hope is that the pretrained model will be able to learn from those to more quickly adapt to new environments. ICRL has several desired advantages over Meta-RL. Firstly, the model does not need to be finetuned which is much more computationally expensive than inferencing. Secondly, Meta-RL typically requires many episodes within the new environment to adapt, whereas ICRL has the possibility to learn from a few trajectories, and in our experiments section we show this to be true.

A crucial challenge we identify is the scarcity of task diversity in current RL environments. Our preliminary investigations revealed that simple modifications in environments such as MuJoCo, such as altering limb lengths and weights or setting different goal velocities, fall short in providing sufficient task diversity for learning new behaviors when prompted. This observation has driven us to develop a new RL environment with a vastly greater range of tasks. Our environment is designed to probe the question, "What enables in-context reinforcement learning?" Through extensive experimentation, we observe significant factors that lead to the emergence of non-Bayesian learning in RL settings. Alongside increasing task diversity, we also explore additional dimensions that contribute to this phenomenon, including model architecture and regularization techniques.

A critical discovery of our study is the intricate relationship between task diversity and the performance of transformers in RL settings. *When the task diversity is limited, transformers tend to emulate a form of Bayesian inference or posterior sampling based predominantly on the tasks they have encountered.* This approach, while effective in *known* scenarios, leads to suboptimal outcomes when dealing with out-of-distribution, unseen tasks. However, our research reveals a significant shift in this pattern as we increase task diversity. With a broader spectrum of tasks during the pretraining phase, transformers demonstrate remarkable improvements in handling unseen tasks, exhibiting enhanced adaptability and efficiency. This observation is not only pivotal in understanding transformer behavior in RL environments but also illuminates a new pathway for designing RL environments. By prioritizing task diversity, we can develop environments that nurture more robust and versatile agents, capable of superior performance in a wide array of scenarios.

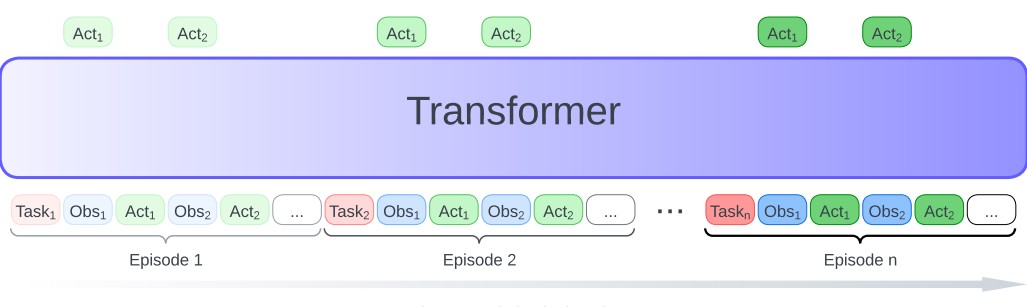

Figure 1: The transformer model is pretrained on multiple episodes autoregressively only for the actions. In each observation, the model predicts the next action in that episode. Rewards for the previous timesteps are included in the observations.

## 2 RELATED WORK

**Meta-reinforcement learning**    The aim of meta-learning, also referred to as learning-to-learn, is designing techniques that yield models that can quickly adapt to unseen tasks (Schmidhuber, 1987; Finn et al., 2017). In reinforcement learning, new tasks can differ by transition probabilities, reward functions, state-action spaces or constraints (Mitchell et al., 2021; Zintgraf et al., 2021; Khattar et al., 2022). The possible ways of adapting meta models include finetuning (Gupta et al., 2018; Sel et al., 2023b; Padalkar et al., 2023), using regressive models to extract latent features from demonstrations (Zintgraf et al., 2019; Dorfman et al., 2021; Gehring et al., 2022; Khattar & Jin, 2023) or Bayesian inference (Humplik et al., 2019; Rothfuss et al., 2021).

**In-context learning.**    ICL is a form of meta-learning more attributed to autoregressive models such as RNNs (Rumelhart et al., 1986; Hochreiter & Schmidhuber, 1997) and transformers (Vaswani et al., 2017) underlining the parameter-update-free version of meta-learning. These models, when trained on sufficiently diverse and large datasets, can learn new tasks only by being provided examples into their context (Brown et al., 2020; Wei et al., 2022; Sel et al., 2023a). Transformers can also portray the ability to infer tasks when prompted with trajectories (Laskin et al., 2022; Xu et al., 2022; Lee et al., 2023). In this paper, we investigate the emergence of in-context out-of-distribution reinforcement learning through autoregressive pretraining.

## 3 IN-CONTEXT RL SETTING

We focus on some task distribution $P_{\text{pre}}$ on an infinite set of finite-horizon Markov decision processes (MDP) $\mathcal{T}$. Each task $\mathcal{T} \in \mathcal{T}$ is defined by the tuple $(\mathcal{S}, \mathcal{A}, T_{\mathcal{T}}, R_{\mathcal{T}}, \rho_{\mathcal{T}}, H)$, where $\mathcal{S}$ denotes the state space, $\mathcal{A}$ is the action space, the transition function $T : \mathcal{S} \times \mathcal{A} \to \Delta(\mathcal{S})$, the reward function $R : \mathcal{S} \times \mathcal{A} \to \Delta(\mathbb{R})$, $\rho$ is the initial state distribution and $H$ represents the finite horizon. For any MDP $\mathcal{T}$, we denote its optimal policy by $\pi_{\mathcal{T}} : \mathcal{S} \to \Delta(\mathcal{A})$. In the rest of the paper, we use tasks and MDPs interchangeably.

### 3.1 PRETRAINING.

Let $\phi$ be a distribution on tasks $\mathcal{T}$. The training dataset distribution $\mathcal{D}_{\text{pre}}^{\phi}$ consists of $n$ concatenated trajectories $D = (\tau^1, \dots, \tau^n)$, where each trajectory $\tau^k = \{(s_j, a_j, r_j, s'_j)\}_{j=0}^{H-1}$ is sampled in task $\mathcal{T} \sim \phi$ with its optimal policy $\pi_{\mathcal{T}}$ for $k \in [n]$. Also let $H_k^D = \{(s_j, a_j, r_j, s'_j)\}_{j=0}^{k}$ denote the partial timesteps on the collection of $n$ trajectories $D$, where $k \in [nH-1]$ with $H_{-1} = \varnothing$.

We represent the transformer model as $M$ that take in the current state as well as the previous timesteps to give a distribution over the action space. We train its parametrization $M_\theta$ on $\mathcal{D}_{\text{pre}}^{\phi}$, where $\theta \in \Theta$ and $\Theta$ is the set of possible parameters, e.g., transformer weights:

$$\min_{\theta \in \Theta} \mathbb{E}_{D \sim \mathcal{D}_{\text{pre}}^{\phi}(\cdot)} \sum_{(H_{j-1}, s_j, a_j) \in D} \ell(M_\theta(\cdot | H_{j-1}, s_j), a_j), \tag{1}$$

where $\ell$ can be chosen to be the log likelihood loss function, $-\log(M_\theta(a_j|H_{j-1}, s_j))$. Now, we investigate two cases: true-diversity pretraining with $\phi = P_{\text{pre}}$ and finite-diversity pretraining with $\phi : \mathcal{X} \to [0, 1]$ where $\mathcal{X} \subseteq \mathcal{T}$ and finite.

### 3.1.1 TRUE-DIVERSITY PRETRAINING

In true-diversity pretraining case, we assume the pretraining dataset task distribution $\phi$ is exactly the pretraining task distribution $P_{\text{pre}}$. Assuming that the pretrained model is consistent, i.e., $\mathcal{D}_{\text{pre}}^\phi(a_j|H_{j-1}, s_j) = M_\theta(a_j|H_{j-1}, s_j)$, we show that the model chooses its actions in a particular state, by posterior sampling from $\mathcal{T}$ according to the partial trajectory in its context and executing optimal policy for the current state.

**Bayesian Inference** Also known as Thompson sampling or posterior sampling, can be used under uncertain in MDPs although first being originated for multi-armed bandits (Thompson, 1933). Briefly, the idea is to maintain a posterior distribution over the tasks, and then act optimally for the task you sample from that distribution. After, observing the new information such as the reward and the next state, the beliefs over the MDPs are updated accordingly.

**Theorem 3.1.** *Assume that we use the log likelihood loss function in the pretraining objective equation 1 over the dataset $\mathcal{D}_{\text{pre}}^{P_{\text{pre}}}$. If the pretrained transformer model $M_\theta$ is consistent, i.e., $\mathcal{D}_{\text{pre}}^{P_{\text{pre}}}(a_j|H_{j-1}, s_j) = M_\theta(a_j|H_{j-1}, s_j)$, then we have*

$$P(a_{\text{ps}} = a|H_{t-1}, s_t) = M_\theta(a|H_{t-1}, s_t), \tag{2}$$

*for all $a \in \mathcal{A}$, and for all $H_{t-1}$, $s_t$ generated in some task $\mathcal{T} \sim P_{\text{pre}}(\cdot)$ by unrolling its optimal policy, where $a_{\text{ps}}$ is the action chosen according to optimal policy for the sampled MDP from the updated belief distribution according to the partial trajectory $H_{t-1}$.*

Theorem 3.1 shows that this pretraining setup will result in a Bayesian inference decision making when the pretraining dataset is generated by the exact pretraining task distribution. This is encouraging, because the transformer model can learn to reason under the uncertainty of the current MDP it is in, only by supervised pretraining on the true task diversity.

### 3.1.2 FINITE-DIVERSITY PRETRAINING

We are also interested in the case where we have limited number of tasks, which is much more common in practical scenarios. Let us update our pretraining dataset $\mathcal{D}_{\text{pre}}^\phi$ by assuming that it is generated in $N$ tasks $\mathcal{X} = \mathcal{T}_1, \cdots, \mathcal{T}_N$ and their optimal policies $\pi_1, \ldots, \pi_N$, respectively. Then $\phi$ is a distribution over this finite set $\mathcal{X}$. This time it is not straightforward to know how the model would behave when prompted with context from tasks that do not appear in $\mathcal{D}_{\text{pre}}$. One possible option is Bayesian inference on this limited number of tasks:

$$M_\theta^{\text{F-PS}}(a|H_{t-1}, s_t) = \sum_{i=1}^N \pi_i(a|s_t) \cdot \frac{\prod_{j=0}^{t-1} \pi_i(a_j|s_j)R_i(r_j|s_j, a_j)T_i(s_j'|s_j, a_j)}{\sum_{k=1}^N \prod_{j=0}^{t-1} \pi_k(a_j|s_j)R_k(r_j|s_j, a_j)T_k(s_j'|s_j, a_j)}, \tag{3}$$

for all $a \in \mathcal{A}$ and any task $\mathcal{T}$. Despite this model being able to identify the current task if it is seen during pretraining given enough trajectories, it can still be arbitrarily bad for a general set of tasks $\mathcal{T}$. We see in Figure 2 that when the task diversity is low, the pretrained transformer model chooses its actions very similarly to equation 3.

A more effective strategy is posterior sampling with an estimated prior $\hat{P}_{\text{pre}}$, that is non-zero for any task $\mathcal{T}$ if $P_{\text{pre}}(\mathcal{T})$ is also non-zero:

$$M_\theta^{\text{E-PS}}(a|H_{t-1}, s_t) = \int_{\mathcal{T}' \in \mathcal{T}} \pi_{\mathcal{T}'}(a|s_t)\hat{g}(\mathcal{T}'|H_{t-1})d\mathcal{T}', \tag{4}$$

where $\hat{g}$ is the posterior over $\mathcal{T}$ after observing $H_{t-1}$:

$$\hat{g}(\mathcal{T}|H_{t-1}) = \frac{\prod_{j=0}^{t-1} \pi_{\mathcal{T}}(a_j|s_j)R_{\mathcal{T}}(r_j|s_j, a_j)T_{\mathcal{T}}(s_j'|s_j, a_j)\hat{P}_{\text{pre}}(\mathcal{T})}{\int_{\mathcal{T}' \in \mathcal{T}} \prod_{j=0}^{t-1} \pi_{\mathcal{T}'}(a_j|s_j)R_{\mathcal{T}'}(r_j|s_j, a_j)T_{\mathcal{T}'}(s_j'|s_j, a_j)\hat{P}_{\text{pre}}(\mathcal{T}')d\mathcal{T}'}. \tag{5}$$

A notable aspect of this strategy is its adaptability to the quality of policies encountered during testing. Even when presented with trajectories from suboptimal policies, the model can leverage environmental cues—such as reward signals and transition dynamics—to discern the underlying task structure and infer the optimal policy. This result is distinguished from pure imitation learning paradigms. Instead, it embodies a form of reinforcement learning where the pretrained transformer has the potential to surpass the performance of the demonstrator policy used for prompting. However, we want to underline that the prompted policy is still important, and it being optimal or close to optimal results in better task recognition, since under this few-shot setting, only rewards or transition dynamics are not enough to learn quickly. This

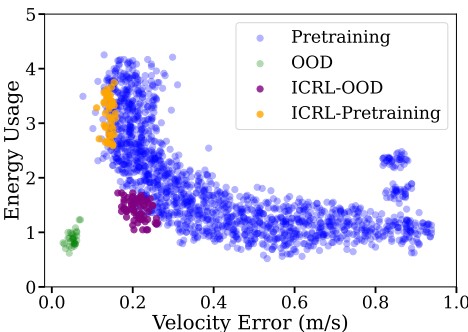

Figure 2: Transformer trained autoregressively on policy trajectories to run at 1 m/s. When prompted with out-of-distribution data (ICRL-OOD), it performs Bayesian Inference only on seen tasks.

phenomena is also dominant in language pretrained transformers (Min et al., 2022). In the remainder of this section, we analyze the in-context reinforcement learning behaviors of $M_\theta^{\mathrm{F-PS}}$ and $M_\theta^{\mathrm{E-PS}}$.

## 3.2 REGRET IMPLICATIONS

A natural choice for the performance of an RL policy $\mu$ is total reward difference between the optimal policy and itself on task $\mathcal{T}$, which can be denoted by the single-episode regret $\mathcal{R}_\mathcal{T}(\mu)$:

$$\mathcal{R}_\mathcal{T}(\mu) \coloneqq \mathbb{E}_{s_0 \sim \rho_\mathcal{T}}[V_{\mathcal{T},0}^*(s_0) - V_{\mathcal{T},0}^\mu(s_0)], \tag{6}$$

where $V_{\mathcal{T},0}^*(s)$ and $V_{\mathcal{T},0}^\pi(s)$ are the expected value functions at state $s$ of task $\mathcal{T}$ at the first time step of the optimal policy and $\mu$, respectively.

**Expected $n$-th episode regret.** Since the models $M_\theta^{\mathrm{F-PS}}$ and $M_\theta^{\mathrm{E-PS}}$ can be prompted with trajectories, we can also consider the expected single-episode regret over the pretraining tasks after being prompted with some number of trajectories, which we denote by $\mathcal{R}_\mathcal{T}^n$:

$$\mathcal{R}_\mathcal{T}^n(\pi) = \mathbb{E}_{\mathcal{T} \sim P_{\mathrm{pre}}} \mathbb{E}_{\boldsymbol{\tau}^n \sim \mathcal{D}_{\mathrm{pre}}(\cdot|\mathcal{T})} \mathbb{E}_{s_0 \sim \rho_\mathcal{T}}[V_{\mathcal{T},0}^*(s_0) - V_{\mathcal{T},0}^{\pi(\boldsymbol{\tau}^n)}(s_0)], \tag{7}$$

where $\boldsymbol{\tau}^n = (\tau^1, \ldots, \tau^n)$. Now we can establish the relation between the finite posterior sampling and the posterior sampling with estimated prior with the following result.

**Theorem 3.2.** *Assume that $|R_\mathcal{T}(s,a)| \leq r_{\max}$ for $s$, $a$ and $\mathcal{T}$. Then, the relation between the average $n$-th episode regret of $M_\theta^{\mathrm{F-PS}}$ and $M_\theta^{\mathrm{E-PS}}$ for the worst family of tasks $\boldsymbol{\mathcal{T}}$ is:*

$$\mathcal{R}_{\boldsymbol{\mathcal{T}}}^n(M_\theta^{\mathrm{E-PS}}) \leq O\left(\frac{H|\mathcal{S}|r_{\max}\sqrt{|\mathcal{A}|\log(|\mathcal{S}||\mathcal{A}|)}}{\sqrt{n}}\right) \leq 2Hr_{\max} \leq \mathcal{R}_{\boldsymbol{\mathcal{T}}}^n(M_\theta^{\mathrm{F-PS}}). \tag{8}$$

The result in Theorem 3.2 indicates that posterior sampling on the finite pretraining tasks can exhibit arbitrarily bad performance, due to only being able to represent some weighted version of the pretraining policies. When presented with a new task, this estimation can lead to suboptimalities that cannot be handled by increasing the in-context trajectories. On the other hand, posterior sampling with an estimated prior covering the task space, does benefit from increased in-context examples. Furthermore, this result is independent on the quality of the prior at estimating the true pretraining distribution.

In cases where the test task appears in the pretraining tasks, we can have a similar result to Theorem 3.2 for $M_\theta^{\mathrm{F-PS}}$ as well. This can be easily seen by setting the true tasks $\boldsymbol{\mathcal{T}}$ to the tasks in the pretraining dataset. Then, it directly follows that finite posterior sampling is efficient in this case. Since the posterior sampling improves with more examples, we can also bound that worst case performance difference between two methods.

**Corollary 3.3.** *Assume that task $\mathcal{T}$ is in the pretraining of $M_\theta^{\mathrm{F-PS}}$. Then, the $n$-th episode expected performance between $M_\theta^{\mathrm{F-PS}}$ and $M_\theta^{\mathrm{E-PS}}$ can be bounded as:*

$$\mathbb{E}_{s_0 \sim \rho_\mathcal{T}} \left[ V_{\mathcal{T},0}^{M_\theta^{\mathrm{E-PS}}} \right] \geq \mathbb{E}_{s_0 \sim \rho_\mathcal{T}} \left[ V_{\mathcal{T},0}^{M_\theta^{\mathrm{F-PS}}} \right] - O\left( \frac{H|\mathcal{S}|\sqrt{|\mathcal{A}|\log(|\mathcal{S}||\mathcal{A}|)}}{\sqrt{n}} \right). \tag{9}$$

# 4 EXPERIMENTS

In this section, we embark on a comprehensive exploration to investigate the impact of pretraining task diversity on the generalization capabilities of transformers in reinforcement learning (RL) environments, especially concerning their adaptability to tasks not encountered during training. This inquiry is crucial for advancing our understanding of in-context learning (ICL) within the domain of RL.

Our experimental framework is designed to address a fundamental question: "How does increasing task diversity in RL environments enable transformers to surpass the limitations of traditional learning paradigms?" To explore this, we recognized the necessity of an RL environment capable of presenting a vast array of distinct tasks and allowing control over their diversity. However, our attempts to find or adapt existing RL environments such as MuJoCo (Todorov et al., 2012), following a number of prior meta-RL works that studied quick adaptation in similar settings (Nagabandi et al., 2018; Rakelly et al., 2019; Yu et al., 2020; Pong et al., 2022), proved inadequate. While these environments permit the creation of numerous variations by minor modifications in reward functions or agent model configurations, they fall short in generating the level of task diversity required for our study, leading to suboptimal generalization in unseen tasks during testing (see Figure 2).

Acknowledging this gap, we repurposed the Omniglot dataset, primarily utilized in supervised learning tasks, to develop a novel RL benchmark. This benchmark is tailored to provide the extensive task variety essential for our question, enabling us to thoroughly examine how task diversity influences transformers' ability to learn and generalize beyond their training experiences. Our experiments delve into multiple facets of this phenomenon. We meticulously evaluate how transformers, traditionally bounded by Bayesian inference, evolve in their learning capabilities as they are exposed to an increasingly diverse range of tasks. This is akin to showing the transition from $M_\theta^{\mathrm{F-PS}}$ to $M_\theta^{\mathrm{E-PS}}$, where the latter might have been seen as unrealistic. We also investigate the effects of model architecture, regularization techniques, and other relevant factors on this learning process. The effect of action representation, such as continuous or discrete inputs is explored in Appendix B.1 due to space constraints.

## 4.1 NOVEL ICRL ENVIRONMENT

The Omniglot dataset (Lake et al., 2019) serves as the foundation for our novel in-context reinforcement learning (ICRL) environment. This dataset is a rich collection of over 19,000 handwritten character images derived from a diverse range of alphabets and contributors. A unique feature of Omniglot is the inclusion of 'strokes' data, which provides a sequential representation of how each character is written. Leveraging this aspect, we have transformed Omniglot into an innovative benchmark for few-shot ICRL.

In our setup, we select $n$ examples from each character along with their corresponding stroke sequences. These are then concatenated to form the input for our transformer model. The primary task in this environment is to learn the writing of a character based on provided demonstrations. The action space for the agent involves deciding where to place strokes on a canvas decided by the task transition dynamics, and the reward function is defined as the negative Euclidean distance from the agent's action to the ground truth stroke placement. Therefore, it is not possible for an agent to write a character correctly when they only view the goal state. However, with provided demonstrations, the learner can understand the transition dynamics together with the exact stroke sequence to correctly write any new character. A key aspect of our environment is the control over task diversity, which we manage by selecting specific characters for inclusion. As we increase the variety of characters, the diversity of pretraining tasks correspondingly expands. This approach allows us to systematically investigate how varying levels of task diversity impact the in-context learning abilities of our models.

We assess the in-context learning capacity of the models by examining their performance on holdout classes—character sets that were not seen during training, directly taken from the evaluation split of the Omniglot dataset.

**Trajectory representation.** The task description, which is the goal image in our environment, is added as a prefix to the time step inputs to the transformer as shown in Figure 1.

## 4.2 MODEL ARCHITECTURE

We use a GPT-2 transformer (Radford et al., 2019) with head and embedding layers removed. Additionally, for processing images of handwritten characters, we finetune a pretrained ResNet model (He et al., 2016). Our approach utilizes projection matrices for both the input and output actions to match the dimensions of these different modalities, a technique employed commonly (Liu et al., 2024). We show these parts in Figure 1, together with the trajectories are represented.

**Image Embeddings.** Our methodology incorporates a finetuned, pretrained ResNet model with configurations of 18, 34, and 50 layers (He et al., 2016). The original pre-trained model lacked the precision needed for accurately embedding the specific locations of strokes within the images. To address this, we adopted a scaled-down version of our training setup. This involved only four inputs to the entire model, consisting of two one-step horizon trajectories with corresponding images and a random point within the strokes, along with another image matched to the same corresponding image. Through this fine-tuning process, the ResNet model effectively learns to identify and extract stroke locations from the images.

**Action Embeddings.** Despite the prevalence of individual tokenization of action dimensions in existing literature (Janner et al., 2021; Brohan et al., 2022), we found no significant advantage in applying this technique to our framework (see Figure 9). Instead, we implement a straightforward projection layer to acquire the action embeddings. This involves the use of a single-layer perceptron without an activation function.

**Transformer.** For our transformer model, we select a GPT-2 architecture (Radford et al., 2019) and modify it by incorporating 4 to 16 layers with embedding dimensions ranging from 16 to 1024. We exclude the original embedding and token head layers and introduce projection layers tailored for image embeddings, input actions, and output actions. A decision was made against utilizing a pre-trained GPT-2 model. The primary reason is the unavailability of pre-trained GPT-2 models in the various layer and embedding dimension formats required for our study.

**Action heads.** We again employ a single-layer perceptron, this time to down-project the final embedding produced by the transformer to derive the actions. The Mean Squared Error (MSE) loss is used, as it allows us to directly obtain continuous actions without the need for tokenization per each action dimension. Other possible choices could include per action dimension outputs (Janner et al., 2021) or diffusion action heads (Chi et al., 2023).

## 4.3 TRAINING DETAILS

Unless specified otherwise, we utilize the AdamW optimizer (Loshchilov & Hutter, 2017) combined with a cosine learning decay schedule (Loshchilov & Hutter, 2016). This schedule includes 1000 warmup steps and reaches a maximum learning rate of 1e-4 over 50,000 training steps. Each sequence incorporates two episodes. For fine-tuning the ResNet models, we employ the AdamW optimizer with a constant learning rate of 1e-5 over 10,000 training steps. While a single consumer-grade GPU with 10 GB of VRAM suffices for the training of all models mentioned (albeit by using gradient accumulation, we expedited our experiments using an 8x H100 node.

## 4.4 RESULTS

For the rest of this section, we provide extensive set of results with various ablation studies to investigate the effects of each important components to in-context reinforcement learning.

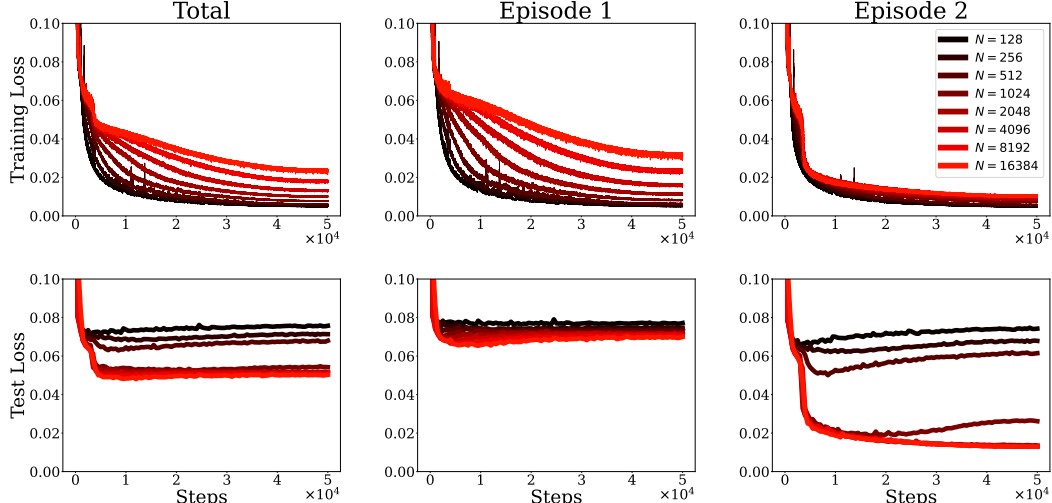

Figure 3: Training loss (top) and testing loss (bottom) for GPT-2 transformer with a layer count of 8 and embedding size of 1024. The episode 2 performance represents the in-context learning capability of the model, by transferring knowledge from episode 1.

#### 4.4.1 EFFECT OF TASK DIVERSITY FOR ICRL

To investigate the effect of task diversity, we varied the number of tasks (64, 128, ..., 16384) while maintaining a constant layer count and embedding size. We also evaluate the model in fixed intervals on the holdout tasks. We provide the overall training loss during training for both training and test tasks in Figure 3. In addition, to more clearly observe the transfer to zero-shot and one-shot setting, we also give the episodic performance for the first and the second episodes. For tasks up to $N = 512$, we see hints of in-context learning up to around 5,000-8,000 gradient steps, after which training loss continues to drop while test loss steadily increases until the end of training. At $N = 1024$, we see a sharp change in in-context learning capability of the model although it is still prone to overfitting by further training. Starting with $N = 2048$ and on, test loss improves throughout the training together with the training.

Zero-shot performance shown in the *first episode* in Figure 3, indicates small gains as we increase the number of tasks. This is due to the nature of the environment, where the the task description is not enough to get a good reward in the environment while still being integral part for the task. Together with the previous episode, the goal of episode 2 is enough to perform the task as seen from the significantly better performance seen in the one-shot setting.

In-context learning is also observed in the training tasks and the performance difference between the zero-shot and one-shot settings increase as more tasks are considered during pretraining. This can be more attributed to less overfitting in zero-shot as one-shot performances are very close regardless of the task count.

#### 4.4.2 VISUALIZATION OF THE TRANSITION FROM BAYESIAN INFERENCE

We qualitatively show the transition from Bayesian inference as task diversity increases in extensive detail in Appendix C, which we cannot show those visualizations in the main paper due to space constraints.

#### 4.4.3 EFFECT OF MODEL CAPACITY

In this section, we focus on the dimensions of transformer architecture, specifically the number of layers and the size of embeddings, as well as the depth of ResNet models used to process image states. For image tokenization via ResNet models, we engage with structures pretrained on ImageNet (Deng et al., 2009), featuring 18, 34, and 50 layers to gauge the influence of depth.

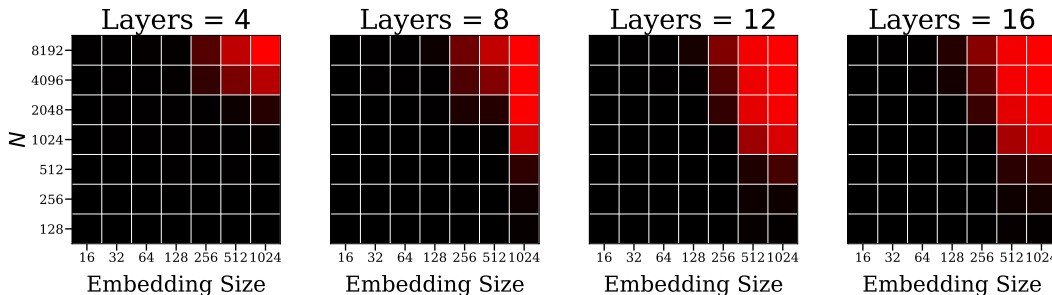

Figure 4: Performance in a new RL environment on unseen tasks, based on the first episode prompts, across models trained on different task counts. Darker shades indicate limited in-context learning, while brighter reds reflect higher generalization ability. Detailed methodology is provided in the appendix.

The performance trends, as depicted in the Figure 4, reveal a clear pattern: an escalation in layer count and embedding size is positively associated with ICL emergence. The 4-layer transformer necessitates the highest task diversity ($N = 8192$) and the largest embedding size before ICL behavior surfaces. With an 8-layer framework, the task diversity threshold for ICL is reduced to $N = 2048$, yet this occurs only at the peak embedding size.

When the architecture is expanded to a 12-layer model, ICL is detectable at a lower task diversity of $N = 1024$, marking a notable improvement. Moreover, this model is capable of ICL at a reduced embedding size of 512, given a task diversity of $N = 2048$. Beyond this point, the expansion to 16 layers fails to produce a proportional decrease in the task diversity requirement for ICL, suggesting a ceiling effect where additional layers offer minimal benefit as seen in Figure 4.

The larger models that possess ICL at test time and the smaller model that do not have similar ICL capabilities during training as shown in Figure 5. This behavior is also seen in large language model pretraining (Chowdhery et al., 2023; Driess et al., 2023). Furthermore, we also observe that *8-layer transformer is performing worse than the 6-layer transformer*. This can be attributed to potential overfitting when pretraining task counts are not the highest in our environment. However, as we increase the layer counts further, we see that models are able to obtain generalization. We believe this hints at a potential double descent phenomenon.

Throughout the models tested, we observe that no model demonstrates ICL with an embedding size smaller than 128. This finding underscores the pivotal role of embedding size. Even with the minimal complexity of a 4-layer model, ICL is attainable with an embedding size of 1024. This indicates that embedding size, rather than layer count, is a more significant factor for achieving ICL in this task.

The analysis of the ResNet models tells a parallel story as shown in Figure 6. The pretrained models, while proficient on a general dataset like ImageNet, were not immediately optimal for our environment's image data. This misalignment necessitated fine-tuning to adjust the models to the specifics of our image data, which includes not just the characters themselves but also the crucial positional information not inherently prioritized in ImageNet training. This is in accordance with the literature hinting at the lack of spatial awareness of vision models simply trained on images and their captions (Gu et al., 2023; Chen et al., 2024).

### 4.4.4 EFFECT OF REGULARIZATION & AUGMENTATIONS

The core of our investigation, as detailed in the accompanying Table 7, revolves around the systematic removal of specific augmentations from the baseline model, which is initially equipped with a full set of augmentations, to gauge their individual contributions to the model's performance on unseen tasks. Our baseline model integrates a comprehensive array of augmentations including image noise, action noise, translation, zoom, rotation, and shearing.

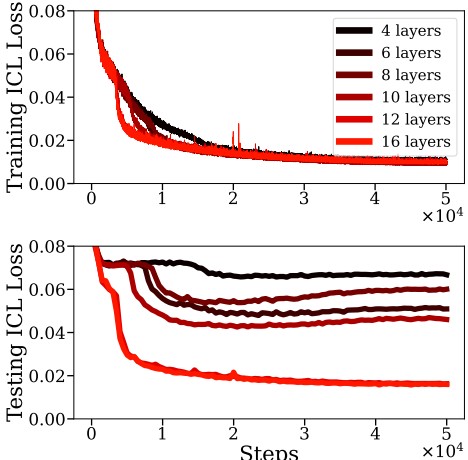

Figure 5: *Training and testing in-context RL performance of transformers with various layer counts.* Task count $N = 2048$ and embedding size is chosen to be 512.

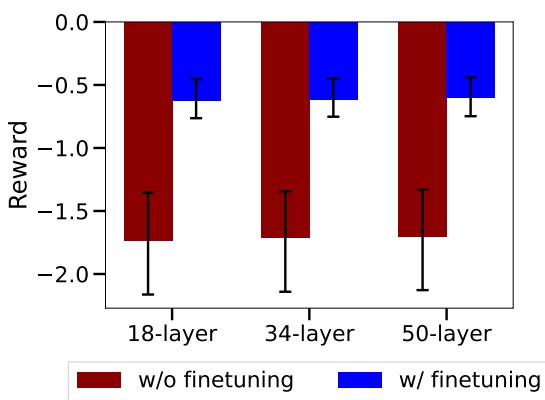

Figure 6: *The effect of finetuning and size of ResNet models to in-context reinforcement learning:* The base transformer model is chosen to be 8-layer GPT-2 with embedding size of 512. The reward is calculated to be the total reward during the second episode in the environment.

| Regularization & Augmentation | Test Loss |
|---|---|
| *Base Model* | $0.13 \pm 0.01$ |
| *- weight decay* | $0.17 \pm 0.01$ |
| *- shear* | $0.25 \pm 0.02$ |
| *- zoom in/out* | $0.29 \pm 0.03$ |
| *- rotation* | $0.43 \pm 0.02$ |
| *- noise* | $0.62 \pm 0.04$ |

Figure 7: Base model is 8-layer GPT-2 with 1024 hidden size trained with the all of the augmentations, weight decay, shear zooming in and out, rotation and addition of noise to images and the actions. The ResNet image embedder is chosen to be the finetuned 18-layer model. We remove each augmentation to see how much performance is lost without it.

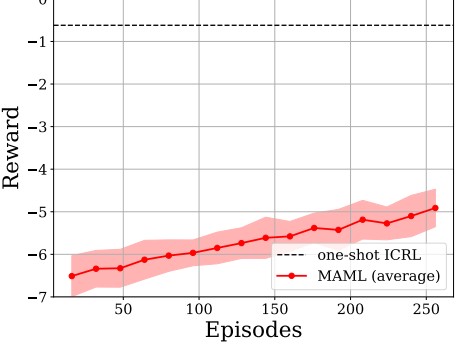

Figure 8: New task adaptation comparison between MAML and One-Shot ICRL.

### 4.4.5 COMPARISON TO MAML

In order to portray the desired advantage of possible quick adaptation that cannot be attained via finetuning on new examples, we trained a MAML agent that inputs the goal state together with the last 5 strokes to output the action for the next stroke on the whole full task diversity setting. For the test (unseen) tasks, we finetuned it for 16 steps with with 16 episodes in each step. As can be seen in Figure 8, there is a significant performance gap compared to one-shot ICRL even after 256 episodes. More details about the training and evaluation is given in Appendix B.2.

## 5 CONCLUSION

In this study, we have examined the factors that lead to generalizable in-context reinforcement learning in transformers. We have introduced a novel RL environment to increase task diversity beyond what is available in any of the other environments we have tested. We have observed that task diversity together with some architectural choices lead way to the emergence in-context learning in unseen tasks. We believe, this work will show the importance of designing RL environments with qualitatively diverse tasks.

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

## A PROOFS

In this section, we provide the omitted proofs in the main text.

### A.1 PROOF OF THEOREM 3.1

**Theorem A.1.** *If the pretrained transformer model $M_\theta$ is consistent, i.e., $M_\theta(a|H_{t-1}, s_t) = \mathcal{D}_{\mathrm{pre}}(a|H_{t-1}, s_t)$, we have*

$$P(a_{\mathrm{ps}} = a|H_{t-1}, s_t) = M_\theta(a|H_{t-1}, s_t), \tag{10}$$

*for all $a \in \mathcal{A}$, and for all $H_{t-1}$ and $s_t$ generated in some task $\mathcal{T} \sim P_{\mathrm{pre}}(\cdot)$ by unrolling its optimal policy.*

*Proof.* Since $M_\theta$ is consistent, we have

$$M_\theta(a|H_{t-1}, s_t) \tag{11}$$

$$= \mathcal{D}_{\mathrm{pre}}(a|H_{t-1}, s_t) \tag{12}$$

$$= \int_{\mathcal{T} \in \boldsymbol{\mathcal{T}}} \pi_{\mathcal{T}}(a|s_t) \mathcal{D}_{\mathrm{pre}}(\mathcal{T}|H_{t-1}, s_t) d\mathcal{T} \tag{13}$$

$$= \int_{\mathcal{T} \in \boldsymbol{\mathcal{T}}} \pi_{\mathcal{T}}(a|s_t) \frac{\mathcal{D}_{\mathrm{pre}}(H_{t-1}, s_t|\mathcal{T}) \mathcal{D}_{\mathrm{pre}}(\mathcal{T})}{\mathcal{D}_{\mathrm{pre}}(H_{t-1}, s_t)} d\mathcal{T} \tag{14}$$

$$= \int_{\mathcal{T} \in \boldsymbol{\mathcal{T}}} \pi_{\mathcal{T}}(a|s_t) \frac{\mathcal{D}_{\mathrm{pre}}(H_{t-1}, s_t|\mathcal{T}) \mathcal{D}_{\mathrm{pre}}(\mathcal{T}) d\mathcal{T}}{\int_{\mathcal{T}' \in \boldsymbol{\mathcal{T}}} \prod_{(s_j, a_j, r_j, s_j') \in (H_{t-1}, s_t)} \pi_{\mathcal{T}'}(a_j|s_j) T_{\mathcal{T}'}(s_j'|s_j, a_j) R_{\mathcal{T}'}(r_j|s_j, a_j) \rho_{\mathcal{T}'(s_0)} \mathcal{D}_{\mathrm{pre}}(\mathcal{T}') d\mathcal{T}'} \tag{15}$$

$$= \int_{\mathcal{T} \in \boldsymbol{\mathcal{T}}} \pi_{\mathcal{T}}(a|s_t) \frac{\mathcal{D}_{\mathrm{pre}}(H_{t-1}, s_t|\mathcal{T}) P_{\mathrm{pre}}(\mathcal{T}) d\mathcal{T}}{\int_{\mathcal{T}' \in \boldsymbol{\mathcal{T}}} \prod_{(s_j, a_j, r_j, s_j') \in (H_{t-1}, s_t)} \pi_{\mathcal{T}'}(a_j|s_j) T_{\mathcal{T}'}(s_j'|s_j, a_j) R_{\mathcal{T}'}(r_j|s_j, a_j) \rho_{\mathcal{T}'(s_0)} P_{\mathrm{pre}}/(\mathcal{T}') d\mathcal{T}'} \tag{16}$$

$$= \int_{\mathcal{T} \in \boldsymbol{\mathcal{T}}} \pi_{\mathcal{T}}(a|s_t) \frac{P(H_{t-1}, s_t|\mathcal{T}) P_{\mathrm{pre}}(\mathcal{T})}{P(H_{t-1}, s_t)} d\mathcal{T} \tag{17}$$

$$= \int_{\mathcal{T} \in \boldsymbol{\mathcal{T}}} \pi_{\mathcal{T}}(a|s_t) P(\mathcal{T}_{\mathrm{ps}} = \mathcal{T}|H_{t-1}, s_t) d\mathcal{T} \tag{18}$$

$$= P(a_{\mathrm{ps}} = a|H_{t-1}, s_t). \tag{19}$$

$\square$

### A.2 PROOF OF THEOREM 3.2

**Theorem A.2.** *Assume that $|R_{\mathcal{T}}(s, a)| \leq r_{\max}$ for $s$, $a$ and $\mathcal{T}$. Then, the relation between the expected $n$-th episode regret of $M_\theta^{\mathrm{F-PS}}$ and $M_\theta^{\mathrm{E-PS}}$ for the worst family of tasks $\boldsymbol{\mathcal{T}}$ is:*

$$\mathcal{R}_{\boldsymbol{\mathcal{T}}}^n(M_\theta^{\mathrm{E-PS}}) \leq O\left(\frac{H|\mathcal{S}|r_{\max}\sqrt{|\mathcal{A}|\log(|\mathcal{S}||\mathcal{A}|n)}}{n}\right) \leq 2Hr_{\max} = \mathcal{R}_{\boldsymbol{\mathcal{T}}}^n(M_\theta^{\mathrm{F-PS}}). \tag{20}$$

*Proof.* We start by establishing the last inequality. Since the we consider the worst family of tasks $\mathcal{T}$, we need to take the maximum of such task families that appear in the regret definition:

$$\max_{\mathcal{T}} \mathcal{R}_{\mathcal{T}}^n(M_\theta^{\mathrm{F-PS}}) = \max_{\mathcal{T}} \mathbb{E}_{\mathcal{T} \sim P_{\mathrm{pre}}(\cdot)} \mathbb{E}_{s_o \sim \rho_{\mathcal{T}}} \left[ V_{\mathcal{T},0}^*(s_0) - V_{\mathcal{T},0}^{M_\theta^{\mathrm{F-PS}}}(s_0) \right] \tag{21}$$

$$= \max_{\mathcal{T}} \mathbb{E}_{\mathcal{T} \sim P_{\mathrm{pre}}(\cdot)} \mathbb{E}_{s_o \sim \rho_{\mathcal{T}}} \left[ V_{\mathcal{T},0}^*(s_0) - \min_{\mathcal{T}' \in \{\mathcal{T}_1, \dots, \mathcal{T}_N\}} V_{\mathcal{T},0}^{\mathcal{T}'}(s_0) \right] \tag{22}$$

$$= \max_{\mathcal{T}} \mathbb{E}_{\mathcal{T} \sim P_{\mathrm{pre}}(\cdot)} \mathbb{E}_{s_o \sim \rho_{\mathcal{T}}} \left[ Hr_{\max} - \min_{\mathcal{T}' \in \{\mathcal{T}_1, \dots, \mathcal{T}_N\}} \mathbb{E} \sum_{t=0}^{H-1} r_{\mathcal{T}}(s_t, \pi_{\mathcal{T}'}(s_t)) \right] \tag{23}$$

$$= \mathbb{E}_{\mathcal{T} \sim P_{\mathrm{pre}}(\cdot)} \mathbb{E}_{s_o \sim \rho_{\mathcal{T}}} \left[ Hr_{\max} - \sum_{t=0}^{H-1} -r_{\max} \right] \tag{24}$$

$$= 2Hr_{\max}. \tag{25}$$

We have established that when $M_\theta$ is consistent, it is equivalent to posterior sampling in 3.1. Then, $M_\theta^{\mathrm{E-PS}}$ is a posterior sampling with some estimated priori distribution $\hat{P}_{\mathrm{pre}}$ where $\hat{P}_{\mathrm{pre}}(\mathcal{T})$ is non-zero if $P_{\mathrm{pre}}(\mathcal{T})$ is non-zero due to our assumption. Finally, we can use Theorem 1 on (Osband et al., 2013) to get the first equality. Since $2Hr_{\max}$ is the highest regret possible, we also have the second inequality in the theorem statement.

$\square$

### A.3 PROOF OF COROLLARY 3.3

**Corollary A.3.** *Assume that task $\mathcal{T}$ is in the pretraining of $M_\theta^{\mathrm{F-PS}}$. Then, the $n$-th episode expected performance between $M_\theta^{\mathrm{F-PS}}$ and $M_\theta^{\mathrm{E-PS}}$ can be bounded as:*

$$\mathbb{E}_{s_0 \sim \rho_{\mathcal{T}}} \left[ V_{\mathcal{T},0}^{M_\theta^{\mathrm{E-PS}}} \right] \geq \mathbb{E}_{s_0 \sim \rho_{\mathcal{T}}} \left[ V_{\mathcal{T},0}^{M_\theta^{\mathrm{F-PS}}} \right] - O\left( \frac{H|\mathcal{S}|\sqrt{|\mathcal{A}|\log(|\mathcal{S}||\mathcal{A}|n)}}{n} \right). \tag{26}$$

*Proof.* If the task $\mathcal{T} \in \{\mathcal{T}_1, \dots, \mathcal{T}_N\}$, we can have $M_\theta^{\mathrm{F-PS}}$ be $\pi_{\mathcal{T}}$ given high enough $n$ episodes into its context. Then, its regret would be zero. Using the result from Theorem 3.2, we can directly have the statement in the corollary. $\square$

## B  EXPERIMENT DETAILS

We provide all the hyperparameters used in our training in Table 1.

### B.1  EFFECT OF ACTION REPRESENTATION

We tested the effect of discretizing each action dimension into 256 uniform bins and representing the trajectories as:

$$D = (T_1^1, s_1^1, a_{1,1}^1, a_{1,2}^1, s_2^1, \dots, T_2^2, s_1^2) \sim \mathcal{D}_{\mathrm{pre}}, \tag{27}$$

where $a_{i,j}^k$ is the action token at time step $i$ of dimension $j$ at episode $k$. Since we no longer are using the continuous embeddings directly for the actions, we also removed the input action projection layer, and directly used the embedder of the GPT-2 to learn the embedding. We have chosen the first 256 tokens to represent the action bins. In Figure 9, we plot the minimum loss obtained in the second episode the assess the effect of discretizing actions to ICL performance. However, we do not observe significant difference across a variety of number of tasks.

### B.2  MAML TRAINING DETAILS

For MAML training, we utilize the same architecture for the images together with the concatenated the last 5 strokes data as input to an MLP with 4 layers, each having 1024 neurons. We utilized Adam

| Parameter | Default Value |
|---|---|
| Learning rate | $1 \times 10^{-4}$ |
| Batch size | 1024 |
| Number of iterations | $5 \times 10^4$ |
| Number of warmup steps | 1000 |
| Clip gradient norm | 1.0 |
| Weight decay | 0.0 |
| Image noise value | 10% |
| Stroke noise value | 0.5% |
| Shift value | 10 pixels |
| Zoom value | 30% |
| Rotation value | 360 degrees |
| Shear value | 40% |
| Number of embeddings | 512 |
| Number of layers | 8 |
| Number of heads | 8 |
| Horizon | 50 |

Table 1: Default Hyperparameter Values

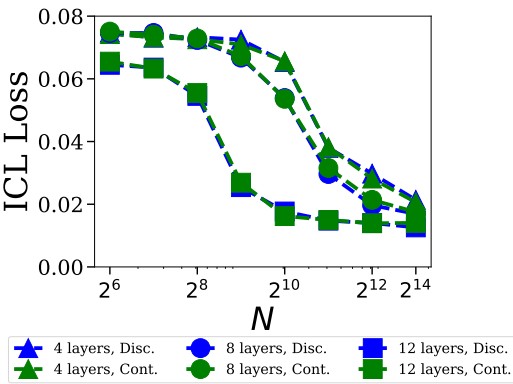

Figure 9: The minimum test loss obtained on the second episode during training. Embedding dimensions are chosen to be 512, and other hyperparameters is chosen to be the default one in our experiments.

optimizer with learning rate 1e-4, batch size 1024 and $5 \times 10^4$ training steps to be comparable to ICRL training. During evaluation, we again simply let the model output the next action, and in the next step we let that action be the last action it has taken as input. The rest of the evaluation for the rewards are the same as before. In Figure 8 we only plot the model trained with MAML that has been trained on the full 16k task diversity. However, we still see a large performance gap showing that finetuning indeed requires more examples compared to an ICRL setup.

## C  THE TRANSITION FROM BAYESIAN INFERENCE

We give the motivation for our proposed RL benchmarks, by visualizing the Bayesian inference on a limited number of tasks in Figure 2 in the MuJoCo HalfCheetah environment (please see the explanation of that figure below). Further, in Figures 3-5 we show this transition from the Bayesian inference on pretraining tasks to a Bayesian inference to a task distribution for the way transformers change their in-context learning method as we increase task diversity by marking the gap in total cumulative reward in training and test tasks.

In order to visualize our main takeaway, we have prompted our models pretrained with task diversity from $N = 2$ to $N = 2048$ similar to the ones shown in Figure 3 in our paper, by the English letters. More specifically, we choose two handwritten characters from each letter in the alphabet. We insert the image and strokes sequence of the first one together with the image of the second one into the context of our pretrained models. Then we let it generate the action sequences. We visualize this in Figures 10-11. As can be seen, when the task diversity is lower than 2048, we see it a trajectory generated a character from its pretraining dataset. In that figure, this is most obvious for $N = 2$, $N = 8$ and $N = 32$ since the trajectories for some of them are similar. However, when $N = 2048$ we see a good representation of the true actions to the ones shown by the expert. We believe this is a clear evidence of the shift in the in-context learning method. The visualizations are also consistent with the Figure 3 in our paper, where we a sudden dropout going from $N = 512$ to $N = 2048$.

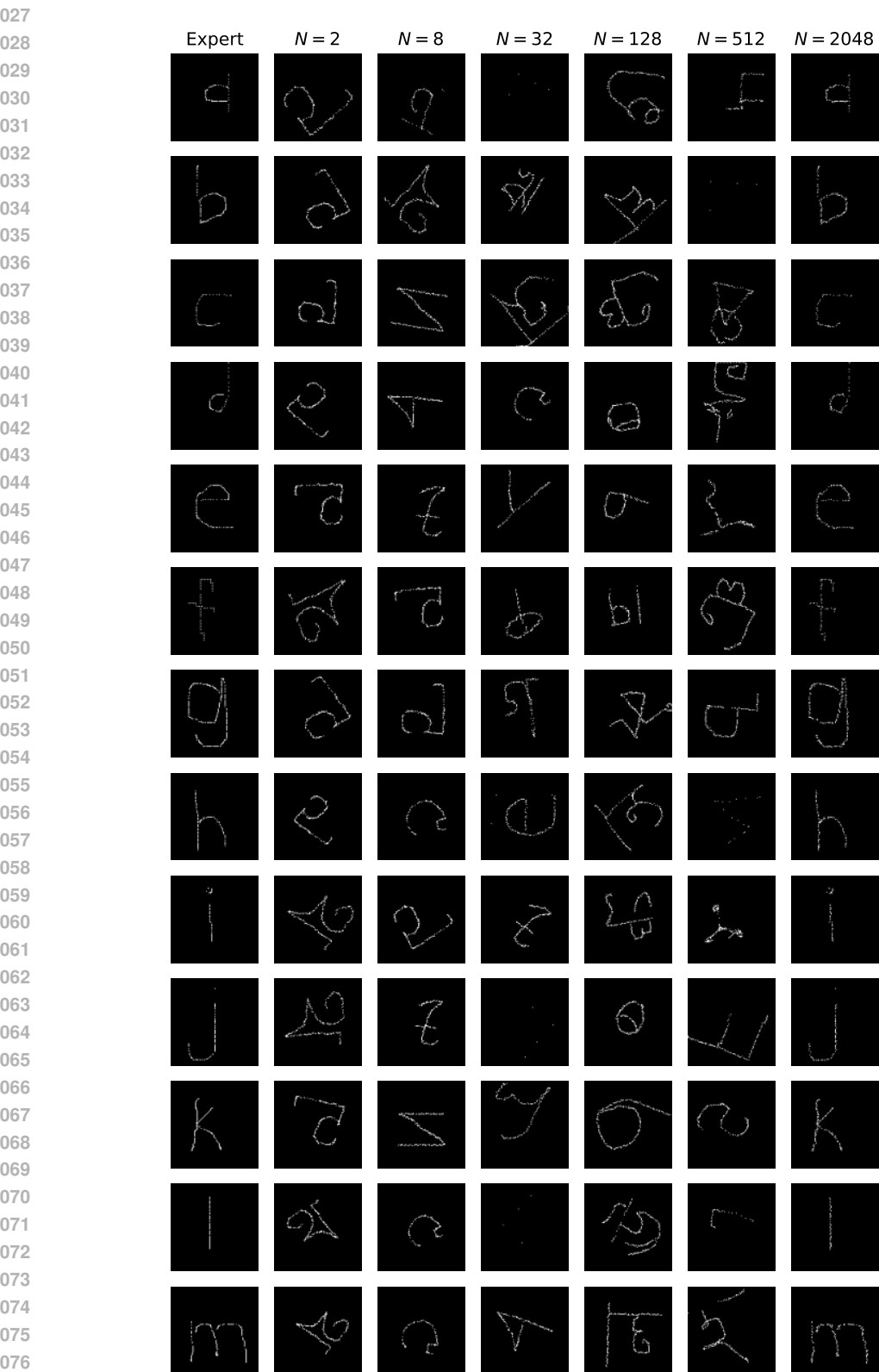

Figure 10: *Shift in in-context RL method as task diversity increases.* Letters from "a" to "m".

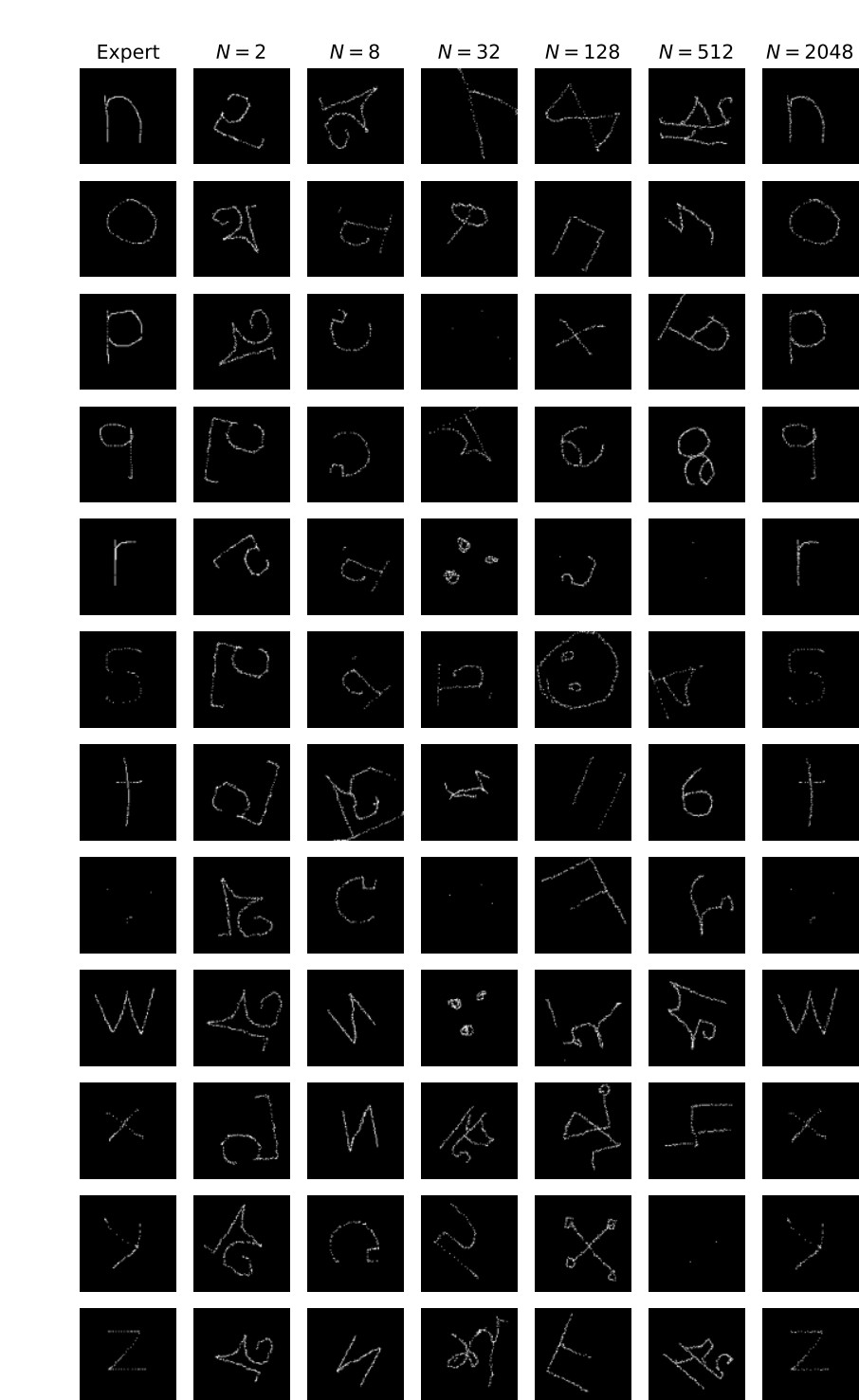

Figure 11: *Shift in in-context RL method as task diversity increases - continued.* Letters from "n" to "z".

