# OpenReview forum: "Actions Speak Louder Than States: Going Beyond Bayesian Inference in In-Context Reinforcement Learning"
_ICLR.cc/2025/Conference — Submitted to ICLR 2025_

### Official Review · Reviewer_WasS · 2024-11-03

**Soundness:** 2
**Presentation:** 1
**Contribution:** 1
**Rating:** 3
**Confidence:** 3

**Summary:**

This paper focuses on in-context RL, where a transformer model has to predict next actions for a pseudo-optimal policy, given the trajectory so far. The transformer is pre-trained on logged data from a optimal policies in a set of MDPs. This work proposes that such architectures can surpass traditional bayesian inference and generalize beyond the MDPs observed at training time. They repurpose the Omniglot dataset to form a new benchmark for evaluating their ideas.

**Strengths:**

* This work focuses on a timely problem regarding in-context RL.
* A new benchmark is created for in-context multi-task reinforcement learning, which the community could benefit from.
* This work includes a detailed regret analysis of two proposed techniques.

**Weaknesses:**

While the premise looks promising, I believe that this paper needs more refinement and revision before it can be helpful.
* I found it difficult to understand what $M_{\theta}^{E-PS}$ looks like in practice. Neither $\mathbfcal{T}$ (the true set of train+test tasks) nor $\hat{g}(\cdot|\cdot)$ (prior/posterior over tasks) seem to be available in practice. Perhaps it would be helpful to include an algorithm for each of $M_{\theta}^{E-PS}$ and $M_{\theta}^{F-PS}$ in the appendix.
* There is a large disconnect between §3.1/3.2 and §4. In §3.1.2, we are introduced to two different action predictors $M_{\theta}^{E-PS}$ and $M_{\theta}^{F-PS}$, and we study their regret behavior in §3.2. But where do these two go in §4? There is no mention, comparison, or even justification for why one might not have been possible. Perhaps the authors should add a few paragraphs to connect these two sections, e.g., how do the findings in §3 help the implementation in §4, etc.

    I would be happy to learn how these are connected, or be notified that this question is already answered in the paper.
* I found it difficult to understand the Omniglot-based benchmark, and had to reread §4.1 several times before I understood it. It would be very helpful to include a visualization of the Omniglot task, even if in the appendix. For example, for 3-5 "trajectories", show what the corresponding states, actions and rewards would be. This visualization is independent of the transformer + ICL based approach that you propose.
* While the comparison to MAML is appreciated, there are other relevant baselines to compare to. To start with, see [1, 2]. As of now, there are few things to learn from the evaluation. The general lesson is that to lower the test loss of this specific technique, we need higher task diversity, larger models and more regularization/augmentation. Unfortunately, these points are not surprising. What would be interesting would be to see that while this specific technique benefits from larger model capacity, baselines do not. Or that more diversity does not change baseline behavior while this technique benefits from it.

[1] Laskin, M., Wang, L., Oh, J., Parisotto, E., Spencer, S., Steigerwald, R., ... & Mnih, V. (2022). In-context reinforcement learning with algorithm distillation. arXiv preprint arXiv:2210.14215.

[2] Sodhani, S., Zhang, A., & Pineau, J. (2021, July). Multi-task reinforcement learning with context-based representations. In International Conference on Machine Learning (pp. 9767-9779). PMLR.

**Questions:**

* If I understand correctly, there is an implicit assumption in §3 that the task identity is latent, correct? If so, a justification would be appreciated, perhaps by mentioning several problem settings and applications where in-context learning with latent task identities makes sense.
* In §3.1.2 and line 196 it is stated that "This time it is not straightforward to know how the model would behave when prompted with context from tasks that do not appear in $\mathcal{D}_{pre}$". Were out-of-distribution tasks ever considered in §3.1.1?
* What is $\mathbfcal{T}$ in Eq. 4? Is it the same $\mathbfcal{T}$ in line 204? Why does the $M_{\theta}^{E-PS}$ predictor have access to it?
* What is $\hat{g}(\cdot|\cdot)$ in Eq. 4? If it is a posterior over the task, where did the prior come from? Shouldn't the regret performance of $M_{\theta}^{E-PS}$ depend on how well the prior matches the true prior?
* Do the descriptions in lines 339-345 relate to the fine-tuning of the ResNet model, or a change in the training methodology for the full stack (ResNet + transformer + RL)?

---

> ### Author Response · Authors · 2024-11-27
> **General Clarifications**
>
> We thank the reviewer for taking the time to review our paper, and as it can be understood from the detailed questions, they have spend a good amount of time on our paper which makes our job much easier. Let us address some key points before responding to specific concerns.
>
> Our paper's primary objective is to investigate the emergence of ICRL in relation to task diversity, rather than proposing a new method or creating the most challenging meta-RL environment. While there may be existing meta-RL environments with high task diversity only promising other alternative we are aware of is [1], we focused on creating an environment that allows for reproducible experiments, feasible offline data generation within academic constraints, and accessibility with limited computational resources.
>
> To our knowledge, this critical topic remains largely unexplored in the literature. Our contribution starts by identifying limitations in current RL environments, introducing a purposefully simple yet effective environment for studying ICRL with task diversity, providing theoretical analysis of observed behaviors, and demonstrating the shift from Bayesian Inference to a more general learning method enabling true ICRL. As far as we are aware, **this work is the only work to demonstrate the latter**.
>
> Regarding our environment's complexity, we used MAML to demonstrate that simple episode-based finetuning cannot achieve optimal performance, confirming the presence of genuine ICRL. While other methods could have been used for this demonstration, this is not the paper's central focus.
>
> [1] Nikulin, Alexander, et al. "XLand-minigrid: Scalable meta-reinforcement learning environments in JAX.”

---

> ### Author Response · Authors · 2024-11-27
> **Responses to Weaknesses & Questions**
>
> **Weaknesses:**
>
> **W1:** They are simply representations, and they enable us to compare the limitations of Bayesian inference on finite number of tasks to the Bayesian inference on a coverage task distribution in terms of regret. These are not representations that we can attain exactly in practice, however, as it is said in Theorem 3.1, using log likelihood for the loss function in equation 1, will result in a Bayesian inference decision making.
>
> **W2:** We thank the reviewer for noticing this possible disconnect. We implied that low task diversity training results in Bayesian inference on finite number of tasks $M_\theta^{\text{F-PS}}$, whereas, high task diversity shifts from this pattern to $M_\theta^{\text{E-PS}}$. We have updated Section 4 to include these points, the changes are in blue text.
>
> **W3:** We understand, however, we still believe the explanation in Section 4.1, is better than any figure we could include. We feel perhaps a short video would help better in understanding, that we can include to the supplementary materials for the camera-ready version.
>
> **W4:** We are not focusing on a new method in this paper, the reason for MAML inclusion is to highlight the "challenge" of the problem setting, not as a comparison. The papers you suggest as baselines, again, do not make sense for the same reason.
>
> **Questions:**
>
> **Q1:** There is an *explicit* assumption in Section 3 that the task identities are latent. We believe it would be difficult to find an ICL setting where the latent task identities are not available, perhaps utilizing a transformer with memory on a single RL task. There are numerous examples we can give. Before that, we would like to point out that the definition of task in RL setting can be very nuanced in real life. For instance, a self-driving car that drives on a wet vs dry road are two different tasks, since different actions will have different rewards, even if the final step observation is the same.
>
> **Q2:** Out-of-distribution tasks cannot be considered, because in that section we are giving theoretical results when the training dataset has the exact distribution for the task distribution.
>
> **Q3:** They are the same. $M^{\text{E-PS}}_\theta$ does not have access to $\boldsymbol{\mathbf{\mathcal{T}}}$. It is possible to have priors for tasks that a model has never seen before. The simplest example is image classification task, where an image has never been included in the training dataset. However, the model still able to give us a distribution over the classes for that same image. The same is also happening here, although the model has never some specific task in the training distribution, it is still possible for that model to have priors for that task.
>
> **Q4:** We really thank you for asking these details when they don't seem right, instead of simply discarding our paper. $\hat{g}(\cdot | \cdot)$ is the posterior over the task definition after observing $H_{t-1}$, as given on line 212. The prior as we denoted as $\hat{P}_{\text{pre}}$, emerges throughout the training, it is not something we specifically train for. In the next parts, we show that as long as what we say in lines 207-208 is true (simply prior being non-zero for all tasks that have non-zero task distribution), our Theorem 3.2 holds. We definitely understand the intuition that the quality of match between the estimated prior and the true prior being important (and it is indeed important), however, in Theorem 3.2, we show that as we increase the context episodes, we can get arbitrarily good regret. For the same $n$ number of episodes, the quality of priors will of course matter.
>
> **Q5:** They only relate to finetuning of the ResNet Model, not the full stack.

---

> > ### Comment · Reviewer_WasS · 2024-11-27
> >
> > I thank the authors for their responses, but regrettably they haven't helped clarify most questions.
> >
> > ---
> >
> > > W1: They are simply representations, and they enable us to compare the limitations of Bayesian inference on finite number of tasks to the Bayesian inference on a coverage task distribution in terms of regret. These are not representations that we can attain exactly in practice, however, as it is said in Theorem 3.1, using log likelihood for the loss function in equation 1, will result in a Bayesian inference decision making.
> >
> > What is "coverage task distribution"? Is there a point to the "finite"-ness of $M_{\theta}^{\text{F-PS}}$, i.e., how would the relationship of $M_{\theta}^{\text{F-PS}}$ and $M_{\theta}^{\text{E-PS}}$ change as $N \rightarrow \infty$?
> >
> > ---
> >
> > > We thank the reviewer for noticing this possible disconnect. We implied that low task diversity training results in Bayesian inference on finite number of tasks $M_{\theta}^{\text{F-PS}}$, whereas, high task diversity shifts from this pattern to $M_{\theta}^{\text{E-PS}}$. We have updated Section 4 to include these points, the changes are in blue text.
> >
> > Does the author refer to the sentence in lines 299-300 in the revision? This is not enough to connect these two separate discussions.
> >
> > ---
> >
> > > We understand, however, we still believe the explanation in Section 4.1, is better than any figure we could include. We feel perhaps a short video would help better in understanding, that we can include to the supplementary materials for the camera-ready version.
> >
> > The contents of the paper should be legible and clear at submission time.
> >
> > ---
> >
> > > There is an explicit assumption in Section 3 that the task identities are latent. We believe it would be difficult to find an ICL setting where the latent task identities are not available, perhaps utilizing a transformer with memory on a single RL task. There are numerous examples we can give. Before that, we would like to point out that the definition of task in RL setting can be very nuanced in real life. For instance, a self-driving car that drives on a wet vs dry road are two different tasks, since different actions will have different rewards, even if the final step observation is the same.
> >
> > Where is it made explicit?
> >
> > > We believe it would be difficult to find an ICL setting where the latent task identities are not available
> >
> > So task identities are available and are not latent? This response is very confusing.
> >
> > > There are numerous examples we can give.
> >
> > Would be appreciated.
> >
> > > Before that, we would like to point out that the definition of task in RL setting can be very nuanced in real life. For instance, a self-driving car that drives on a wet vs dry road are two different tasks, since different actions will have different rewards, even if the final step observation is the same.
> >
> > I'm confused around how this relates to the original question.
> >
> > ---
> >
> > > They are the same. $M_{\theta}^{\text{E-PS}}$ does not have access to $\mathbfcal{T}$. It is possible to have priors for tasks that a model has never seen before. The simplest example is image classification task, where an image has never been included in the training dataset. However, the model still able to give us a distribution over the classes for that same image. The same is also happening here, although the model has never some specific task in the training distribution, it is still possible for that model to have priors for that task.
> >
> > If they are the same, couldn't the prior $\hat{P}\_{\\text{pre}}$ be included in Eq. 3, but still with finite tasks?
> >
> > ---
> >
> > > We really thank you for asking these details when they don't seem right, instead of simply discarding our paper. ...  For the same $n$ number of episodes, the quality of priors will of course matter.
> >
> > Is the regret asymptotic then?

---

> > > ### Author Response · Authors · 2024-12-01
> > >
> > > Thank you for following up. We apologize for creating more confusion. Our initial response was unfortunately written in haste. We sincerely appreciate the reviewer for taking the time to provide this detailed feedback and to engage in the discussion. The comments and questions have helped us identify some inconsistencies in Section 3 regarding $M_\theta^{\text{F-PS}}$ and $M_\theta^{\text{E-PS}}$. Furthermore, we recognize that several sections are still unclear and will use your feedback to improve future revisions of the paper. Thank you again. We have appreciated your feedback throughout the discussion.

---

### Official Review · Reviewer_nrwG · 2024-11-03

**Soundness:** 2
**Presentation:** 2
**Contribution:** 2
**Rating:** 3
**Confidence:** 4

**Summary:**

The paper explores the emergence of in-context reinforcement learning (ICRL) in transformer models, presenting a quantitative analysis on the task variety, model depth and data augmentation techniques required for ICRL to emerge. The authors also propose a new benchmark, which as they claim, surpasses other available datasets and environments in task diversity. Also, authors theoretically show that the performance of the in-context methods is indeed bounded by the diversity of task in the training dataset.

**Strengths:**

- Theoretical contribution: decomposition of pretraining data into two task coverage classes and showing the performance bounds.
- To my knowledge, the first attempt to explore scaling of in-context *reinforcement* learning models, so it is a valuable engineering contribution.
- The paper is generally well-written, the theoretical part is clearly explained.

**Weaknesses:**

1. The paper claims to work in in-context *reinforcement* learning setting. However, the authors prepend the task description (line 328) during the evaluation on the test data.

    On the contrary, the ICRL methods, such as AD [1] or DPT [2], do not precondition on the task. The whole idea of ICRL lies in the adaptation to the uncertainty (i.e. solving unseen tasks) without updating the model weights. In the proposed Omniglot environment, conditioning the model on the final state (task) leaves no uncertainty to adapt to, making the setting goal-conditioned.

2. The main metric authors use for making conclusions about in-context abilities is the L2-loss on the test tasks. This approach is known to be inconclusive [3]. Instead, it is better to report total return over a trajectory, given the work is claimed to be exploring the RL setting.
3. In some places in the paper, the authors report “Reward” (I guess it is actually a return, total sum of rewards on a trajectory). However, it is hard to estimate how well the model solves the tasks based on MSE. It is more common to use mean success rate [1, 2] or some n-th percentile success rate [4]. For the reason it is a challenging task to construct a success rate for such an environment, the Omniglot benchmark may not be a good choice.
4. In Appendix C, the authors claim they show how transformer changes its regime from Bayesian inference by training on increasing number of tasks. However, I am not quite satisfied with this argument. Indeed, when the number of tasks increases, transformer learns to reason in-context, rather than to replicate strokes from its weights. At the same time, this effect has no connection with Bayesian inference, but it is an example of overfitting. Surely, a sufficiently large model can learn to copy a small dataset, but one would not call it Bayesian inference.
5. Authors claim to offer a benchmark with “unprecedented task diversity”, however, the datasets and benchmarks with at least similar diversity exist. For example, XLand [4] (and its open sourced analogue XLand-Minigrid [5] with datasets [6]) offers a tree-like structure of tasks, which is considered challenging.
6. The statement given on line 85 concerning slow adaptation of Meta-RL vs. ICRL is not true.  Сonversely, if one consider AD [1] and VariBAD [7], the latter method requires much less episodes to converge to the optimal solution in Dark Room environment, which highlights the effectiveness of Meta-RL methods over ICRL.

[1] Laskin, Michael, et al. "In-context reinforcement learning with algorithm distillation.”

[2] Lee, Jonathan, et al. "Supervised pretraining can learn in-context reinforcement learning.”

[3] Agarwal, Rishabh, et al. "Many-shot in-context learning.”

[4] Team, Adaptive Agent, et al. "Human-timescale adaptation in an open-ended task space.”

[5] Nikulin, Alexander, et al. "XLand-minigrid: Scalable meta-reinforcement learning environments in JAX.”

[6] Nikulin, Alexander, et al. "XLand-100B: A Large-Scale Multi-Task Dataset for In-Context Reinforcement Learning.”

[7] Zintgraf, Luisa, et al. "Varibad: A very good method for bayes-adaptive deep rl via meta-learning.”

**Questions:**

1. Following Weakness #1 and the statement give on line 317, why do authors consider the task impossible to achieve without seeing the goal? Would not it be enough to learn to follow the reward signal, as it is done in many prior ICRL methods?
2. In the MAML experiment, the model sees the last five strokes, while the transformer contains last 2 **episodes** in its context. Why do authors choose MAML to compare with Meta-RL, given it is not a memory-model? To my mind, the paper would benefit from the comparison with the recent SOTA in Meta-RL: Amago [8].
3. For the main plots (Fig. 3, Fig. 5) the authors report it would be beneficial to see multi-seed averaging. This information would be helpful to access the stability of in-context models.
4. Can the authors release the training hyperparameters in a single table? I understand that there were plenty of experiments, but I would appreciate seeing hyperparameters of the best performing models at least.
5. In AD [1], it is reported that enlarging sequence length can also be beneficial to in-context abilities of transformer. Such an experiment with different sequence sizes would maybe shed some more light on the question of in-context emergence. Comparing tradeoffs between sequence size vs. task diversity is also an interesting detail.
6. I am colorblind, cannot see one graph. Appreciate if in the next revision the style would change.

Although I assess this work as a rejection, I would be happy to discuss the paper during the rebuttal to understand authors’ motivation and then to revise my score.

[8] Grigsby, Jake, Linxi Fan, and Yuke Zhu. "Amago: Scalable in-context reinforcement learning for adaptive agents.”

---

> ### Author Response · Authors · 2024-11-27
> **General Clarifications**
>
> We appreciate your thorough review of our paper. This makes our job much easier. Let us address some key points before responding to specific concerns.
>
> Our paper's primary objective is to investigate the emergence of ICRL in relation to task diversity, rather than proposing a new method or creating the most challenging meta-RL environment. While there may be existing meta-RL environments with high task diversity only promising other alternative we are aware of is [1], we focused on creating an environment that allows for reproducible experiments, feasible offline data generation within academic constraints, and accessibility with limited computational resources.
>
> To our knowledge, this critical topic remains largely unexplored in the literature. Our contribution starts by identifying limitations in current RL environments, introducing a purposefully simple yet effective environment for studying ICRL with task diversity, providing theoretical analysis of observed behaviors, and demonstrating the shift from Bayesian Inference to a more general learning method enabling true ICRL. As far as we are aware, **this work is the only work to demonstrate the latter**.
>
> Regarding our environment's complexity, we used MAML to demonstrate that simple episode-based finetuning cannot achieve optimal performance, confirming the presence of genuine ICRL. While other methods could have been used for this demonstration, this is not the paper's central focus.
>
>
> [1] Nikulin, Alexander, et al. "XLand-minigrid: Scalable meta-reinforcement learning environments in JAX.”

---

> ### Author Response · Authors · 2024-11-27
> **Responses to Weaknesses**
>
> **W1:** For the claim about task preconditioning, specifying a task does not eliminate execution uncertainty. Consider writing the letter 'A': even with this clear goal, significant uncertainties remain about the required force, optimal pen angle, and surface characteristics. If task specification truly eliminated all uncertainty, we would see high rewards/low losses from the first episode in Figure 3, which is not the case.
>
> **W2:** Total trajectory return may not be the most appropriate metric for Meta-RL environments, since episode lengths and reward scales can naturally vary between tasks. An agent could achieve identical performance quality across tasks, yet receive different cumulative rewards simply due to varying episode lengths. In our environment, where episodes can be either short or long, using average reward per step provides a more consistent and meaningful measure of performance that normalizes these duration differences.
>
> **W3:** Success rate is primarily used as a metric in robotics-related RL tasks, not as a standard measurement across RL algorithms. The assumption that Meta-RL benchmarks should be limited to environments that can report success rates is unfounded. Consider OpenAI Gym, one of the most widely used RL environment libraries - none of its environments use success rate as a benchmark metric. These environments are frequently adapted for Meta-RL research by modifying their parameters. This established practice demonstrates that success rate reporting is not a prerequisite for Meta-RL benchmarks.
>
> **W4:** Let's clarify the distinction between overfitting and our observed behavior. Overfitting occurs when a model performs well on training data but poorly on new, similar inputs. This is not what we observe in our results - Figure 3's training losses show consistent performance across new instances of previously seen task types, despite never reusing the exact same instance during training. What we actually observe and theoretically demonstrate in Section 3 is a different phenomenon: strong performance on in-distribution tasks (new instances of familiar task types) but weaker performance on out-of-distribution tasks. This behavior aligns with our theoretical analysis of Bayesian inference under finite task distribution during training. Importantly, once task diversity exceeds a critical threshold, the transformer transitions from operating in a finite Bayesian inference regime to behaving more like $M^{\text{E-PS}}_\theta$.
>
> **W5:** We should clarify our statement regarding "unprecedented task diversity." While you mention a dataset with high task diversity, we note that it was introduced just three months before submission (after May 28th). Following reviewer guidelines, it would be more appropriate to consider this as concurrent work. Moreover, there's a practical consideration regarding dataset accessibility. The size of such datasets often presents significant computational barriers for many researchers. We are curious how much of a computation power would be needed to repeat all our experiments on this dataset. This speaks directly to our goal of creating benchmarks that are accessible within typical academic resource constraints.
>
> **W6:** We believe there has been an issue with naming. We meant Meta-RL methods that require weight updates to the model when we meant "Meta-RL", which is evidenced by our previous sentence in line 84, and we believe the reviewer also understood what we meant. However, we believe the comparison between Ad and VariBAD is still not a direct example of Meta-RL vs ICRL. Both AD and VariBAD are Meta-RL methods that do not require weight updates (finetuning) at test stage (So, they are both ICRL).

---

> ### Author Response · Authors · 2024-11-27
> **Responses to Questions**
>
> **Q1:** The goals are replicating how to "write" a specific character. We cannot write a character without knowing what to write.
>
> **Q2:** Please refer to our clarifications. Briefly, we are not interested in ICRL beating MAML or any other Meta-RL method (btw AMAGO is a scalable way to implement ICRL, they don't do weight updates at test time), we simply want to provide proof that the task is not easy, and the results we get in our main results, are not just some side-effect of simple regulation/capacity increase.
>
> **Q3:** We could certainly do this, but we really don't think it would provide any important information. We provide many runs in different configurations. Does the reviewer see any instability in any one of them?
>
> **Q4:** See already existing Table 1.
>
> **Q5:** We agree, observing more episodes would be interesting to observe. However, we are already observing very good returns after observing a single episode. Therefore, there is no reason to test with more than a single episode as context. Again, this is interesting, and we would also wish to see future works investigating this in possibly other environments.
>
> **Q6:** We believe it would be Figure 6, we updated it. Let us know if you have problems with other figures.

---

> ### Comment · Reviewer_nrwG · 2024-11-27
>
> I thank the authors for their thoughtful rebuttal.
>
> > W1: For the claim about task preconditioning, specifying a task does not eliminate execution uncertainty. [...]
> >
>
> Consider the definition of Meta-RL from [1]:
>
> > Meta-RL is most commonly studied in a problem setting where, given a distribution of tasks, the goal is to learn a policy that is capable of adapting to any new task from the task distribution with as little data as possible"
>
> Following this definition, the setting where the model is conditioned on the goal cannot be viewed from a Meta-RL perspective. I understand that apart from uncertainty in the final goal, there may be some other sources of it (i.e. different physics of the simulator), but I do not spot it in the author's work. Besides, as I mentioned in my review, next-token prediction losses may not be a good evidence of in-context abilities [2], therefore I do not see the results in Figure 3 conclusive enough.
>
> > W2: Total trajectory return may not be the most appropriate metric for Meta-RL environments, since episode lengths and reward scales can naturally vary between tasks. [...]
>
> My point with reference to [2] still holds. In the light of this work, I believe the authors should come up with a more conclusive metric that is not known ambiguous in predicting in-context performance.
>
> Besides, the total return over a trajectory (averaged over tasks) is considered standard in Meta-RL literature [1, 3, 4, 6, 8, 10], so I do not see why this benchmark should deviate from it.
>
> > W3: Success rate is primarily used as a metric in robotics-related RL tasks, not as a standard measurement across RL algorithms. [...]
>
> The success rate metrics or similar to it (counting how many timesteps the agent stands in the goal position after reaching it) are common in current ICRL research [3, 4, 5]. As such, I am not proposing to exclusively use these metrics, but I want to again highlight that MSE is not a good metric for a benchmark, especially for comparing images. I would be happy to know what authors may propose (maybe using a simple cnn that classifies images) for solving this problem.
>
> > W4: Let's clarify the distinction between overfitting and our observed behavior. [...]
>
> I see the point authors make, I am grateful for the clarification.
>
> > W5: We should clarify our statement regarding "unprecedented task diversity." While you mention a dataset with high task diversity, we note that it was introduced just three months before submission (after May 28th).
>
> To be precise here: the authors rightfully notice that datasets which were generated from [5] were indeed released after May 28, but the benchmark itself was published on arxiv 19 Dec 2023. In this light, the claim still sounds too strong, I would advise to ease it a little bit.
>
> > W6 [...] we believe the reviewer also understood what we meant.
>
> Unfortunately, I did not. Please, make the text unambiguous, it is vital to be consistent with generally accepted terminology in the field.
>
> > W6 [...] Both AD and VariBAD are Meta-RL methods that do not require weight updates (finetuning) at test stage (So, they are both ICRL).
>
> However I see the difference between Meta-RL and ICRL minimal, it is important to stick to the terminology. I have never yet seen that VariBAD [6] was called an ICRL method. I understand what the authors try to communicate here (gradient-based Meta-RL [7] vs. context representation Meta-RL [8]). In-context RL (ICRL) as a term was introduced in [3] first [9], where it was proposed to implicitly let the in-context abilities of a transformer to estimate the uncertainty from the history of interactions, rather than explicitly learn the representation of it. Thus, I would refrain from calling VariBAD one of the ICRL method.

---

> ### Comment · Reviewer_nrwG · 2024-11-27
>
> While re-reading the paper, in line 81:
>
> > In ICRL, the demonstrations are expert trajectories and the hope is that the pretrained mode [...]
>
> This is not always the case. For example, in AD [3] (the method that pioneered the field of ICRL) authors use learning histories of single-tasked RL algorithms to extract a policy improvement operator from them. Thus, I would refrain from using such strong propositions.
>
> > Q1: The goals are replicating how to "write" a specific character. We cannot write a character without knowing what to write.
>
> Given there is a signal from the environment, it is possible for the model to understand which character to write. In the aforementioned ICRL and Meta-RL papers the goal location is not known for the agent, but with trial and error (which includes getting the signal from the environment) the model successfully navigates to the right place or chooses the direction for the agent to go.
>
> > Q2: [...] we are not interested in ICRL beating MAML [...] we simply want to provide proof that the task is not easy, and the results we get in our main results, are not just some side-effect of simple regulation/capacity increase.
>
> As I understand, for the main experiments authors use a transformer with 8 layers, 8 heads with 512 hidden dimensions (Table 1), the sequence consists of 2 episodes. For MAML baseline, authors take 4 layers MLP with 1024 hidden dimensions (Appendix B2) and the sequence consists of the last 5 strokes. Roughly, the transformer that authors use should have approximately 30M parameters, while MAML MLP backbone consists of 2.5M parameters at most, not mentioning the transformer's greater sequence length. To my mind, such a comparison is not conclusive or fair.
>
> > Q4 & Q6
>
> Thank you, my concerns are addressed.
>
> ---
>
> Overall, my main concern is that the proposed benchmark is suitable neither for Meta-RL nor for ICRL, since the goal for the model is given as an input and no other uncertainty is left for the model to estimate.
>
> As an example, consider Dark Room environment (one of the most common in ICRL now), but instead of keeping the goal hidden from the model, one trains a simple MLP that takes as an input the position of an agent + a goal grid it has to navigate to. Given enough pre-training examples, this MLP could navigate to any grid (even those that the model did not see during training), but we should not call such a behavior "Meta" (rather, it is called goal-conditioned RL, which is a separate field). The example of such a pre-training method may be found in [5, 11], where authors train a goal-reaching policy to later use it for collecting a dataset of diverse behaviours.
>
> Besides, authors the authors reject to address some of my concerns, i.e.
>
> > We could certainly do this, but we really don't think it would provide any important information.
>
> > Therefore, there is no reason to test with more than a single episode as context.
>
> Considering this, and the fact that my main concern still holds, I am inclined to stand by my score.

---

> ### Comment · Reviewer_nrwG · 2024-11-27
>
> [1] Beck, J., Vuorio, R., Liu, E. Z., Xiong, Z., Zintgraf, L., Finn, C., & Whiteson, S. (2023). A survey of meta-reinforcement learning. arXiv preprint arXiv:2301.08028.
>
> [2] Agarwal, R., Singh, A., Zhang, L. M., Bohnet, B., Rosias, L., Chan, S., ... & Larochelle, H. (2024). Many-shot in-context learning. arXiv preprint arXiv:2404.11018.
>
> [3] Laskin, Michael, et al. "In-context reinforcement learning with algorithm distillation.”
>
> [4] Lee, Jonathan, et al. "Supervised pretraining can learn in-context reinforcement learning.”
>
> [5] Nikulin, Alexander, et al. "XLand-minigrid: Scalable meta-reinforcement learning environments in JAX.”
>
> [6] Zintgraf, L., Shiarlis, K., Igl, M., Schulze, S., Gal, Y., Hofmann, K., & Whiteson, S. (2019). Varibad: A very good method for bayes-adaptive deep rl via meta-learning. arXiv preprint arXiv:1910.08348.
>
> [7] Mehta, B., Deleu, T., Raparthy, S. C., Pal, C. J., & Paull, L. (2020). Curriculum in gradient-based meta-reinforcement learning. arXiv preprint arXiv:2002.07956.
>
> [8] Rakelly, K., Zhou, A., Finn, C., Levine, S., & Quillen, D. (2019, May). Efficient off-policy meta-reinforcement learning via probabilistic context variables. In International conference on machine learning (pp. 5331-5340). PMLR.
>
> [9] I suggest the authors do a scholar search with a query "in-context reinforcement learning", limiting the results up until 2021 as a proof to my point. AD was released in 2022.
>
> [10] Choshen, E., & Tamar, A. (2023, July). Contrabar: Contrastive bayes-adaptive deep rl. In International Conference on Machine Learning (pp. 6005-6027). PMLR.
>
> [11] Park, S., Frans, K., Eysenbach, B., & Levine, S. (2024). OGBench: Benchmarking Offline Goal-Conditioned RL. arXiv preprint arXiv:2410.20092.

---

> > ### Author Response · Authors · 2024-12-01
> >
> > Thank you for your thorough response. We appreciate the time and effort you have put in to respond and help improve the paper. We would like to revise our previous response to your initial review. Our initial response was written in haste and regrettably did not give your thoughtful critiques the consideration they deserved.
> >
> > **Task uncertainty in Omniglot benchmark**
> >
> > We agree with the reviewer’s distinction between meta-RL and goal-conditioned RL and will strive to make the framing and language consistent with existing literature. Indeed, our setting does not quite reflect the traditional meta-RL setup, as the task uncertainty is mostly resolved through the observation of the goal in our setting. In this setting there is still information gained from prior interactions (e.g., inferring the correct order of strokes) to improve the learner in-context in the next episodes.
> >
> > **Use of demonstrations**
> >
> > Indeed, AD does not make use of any expert demonstrations at test time or training. It may be more precise for us to say that our work operates in a specific ICRL setting that utilizes demonstrations, similar to prior work Prompt-DT (Prompting Decision Transformer for Few-Shot Policy Generalization; Xu, ICML 2022) or in-context imitation learning. We plan to clarify the problem setup in future revisions, and sincerely appreciate the reviewer’s feedback here.
> >
> > **Gradient-based meta-RL versus context-based meta-RL**
> >
> > Yes, you are absolutely right – we were trying to refer to the difference between gradient-based meta-RL versus context-based meta-RL, but we used the wrong terminology. We apologize for creating confusion in the paper and in the rebuttal. We will be sure to correct the language and ensure that it is consistent with the literature in future versions of the paper.
> >
> > **Metrics**
> >
> > We appreciate the feedback on the metrics and acknowledge that MSE may be misleading. We will look into alternative ways to evaluate in future revisions of the paper.
> >
> > Lastly, we sincerely apologize for the dismissive response to your suggestions on averaging over seeds and analyzing the effect of context length. Maintaining rigor and fully exploring possibilities like the ones you suggested are indeed critical for research, and we will certainly use these suggestions to improve the quality of the results. We appreciate your upholding of these standards, and we will aim to incorporate them in the revision.
> >
> > Again, we’d like to thank you for your feedback and engagement in the discussion, and we understand your decision to maintain the score.

---

### Official Review · Reviewer_pHb4 · 2024-11-04

**Soundness:** 3
**Presentation:** 2
**Contribution:** 3
**Rating:** 5
**Confidence:** 3

**Summary:**

This work discusses the factors affecting the generalization capabilities of an In-Context Learning (ICL) approach in the Reinforcement Learning setting. It especially focuses on how transformers can surpass Bayesian inference limitations for ICL. One of the main factors studied in this work is the task diversity. This factor has been studied theoretically and empirically, showing the importance of a diverse set of tasks to train the ICL algorithm. As mentioned, since the diversity of the existing RL environments (e.g., Mujoco) is not enough to show highly diverse tasks, this work proposed a novel benchmark based on the Omniglot dataset, offering unprecedented task diversity. Aside from task diversity, other factors like model capacity (Sec. 4.4.3), regularization techniques, and augmentation strategies (Sec. 4.4.4) have been studied. The empirical results have shown the importance of task diversity when learning an ICL algorithm with transformers.

**Strengths:**

- The problem discussed in this work is important. The motivation for the problem was relatively clear.
- The empirical results were enough to show the importance of task diversity.
- The ablation studies were comprehensive illustrating the other important factors in designing an ICL algorithm in RL based on transformers.

**Weaknesses:**

- The related work section partially covers the related topics. I would kindly ask for more detailed coverage of the role of task diversity for ICL in supervised learning literature, as mentioned briefly in Lines 048-049.
- The illustrated figure (in Fig. 1) is simple and does not include the essential components mentioned in the architectural description (Sec. 4.2).
- It was mentioned that a preliminary study had shown the limitations of existing RL environments in providing sufficient task variety. However, I could find that in the main paper or the appendix. I would kindly ask for such a study to highlight the importance of the proposed ICL benchmark that is based on Omniglot.
- The measure of diversity is not clear in the paper, which is the main factor studied in this work, aside from being the reason why the novel ICL benchmark was introduced. It could be already covered in other works, but this also supports my claim that the related work section is not comprehensive.
- The reward function in Sec. 3 was introduced as a probability function in $\mathbb{R}$ ($R: \mathcal{S} \times \mathcal{A} \to \Delta(\mathbb{R})$). However, in Theorem 3.2, the absolute value of the reward function was directly used ($R_T (s, a)| ≤ r_{max}$).
- In Fig. 4, the plot is not clear. Having a color bar can improve the readability of the plot.

**Questions:**

I have mainly one important question besides some requests I provided in the weaknesses section:
- **What measure of task diversity has been used in this work?** This is crucial to understand the benefit of introducing the novel ICL benchmark and the limitations of the existing RL environments. I understand from the theoretical section that diversity is about covering the true set of pre-training task distribution, but empirically, it is unclear.

---

### Official Review · Reviewer_7Cao · 2024-11-08

**Soundness:** 3
**Presentation:** 3
**Contribution:** 3
**Rating:** 8
**Confidence:** 4

**Summary:**

This paper presents an analysis of In-Context Learning (ICL) in Reinforcement Learning (RL) and introduces a new benchmark based on Omniglot for evaluating ICL in RL. Using this benchmark, the authors demonstrate how factors such as the number of tasks in the pretraining dataset, architectural hyperparameters of the transformer, and regularization techniques influence the ICL capabilities of a GPT-2-based transformer model trained to autoregressively predict the next action.

**Strengths:**

- The analysis of how the number of tasks in the pretraining dataset and model capacity impact ICL abilities in RL provides valuable insights into the effectiveness of framing RL as an autoregressive problem.

**Weaknesses:**

- In addition to the factors considered in the paper, I think it would be beneficial to include an ablation study on the impact of sequence length on ICL abilities for RL.
- See the Questions section for additional clarifications.

**Questions:**

1. What is the specific sequence that is used as input to the transformer?

---

> ### Author Response · Authors · 2024-11-27
> **Rebuttal**
>
> Thank you for taking the time to review our paper. We are really encouraged by your high view of our paper.
>
> **Responses to Questions:**
>
> **Q1**: We agree, observing more episodes and different context lengths would be interesting to explore, especially given observations of positive results from previous studies such as AD. We would also wish to see future works investigating this in possibly other environments.
>
> **Q2**: We supply the previous observations, rewards and the actions in the trajectory.

---

### Meta-Review · Area_Chair_Rrx1 · 2024-12-20

**Metareview:**

This paper tackles an important problem in in-context learning (ICL) for reinforcement learning (RL), demonstrating the impact of task diversity. While the empirical results and ablations are strong, the submission suffers from several weaknesses that prevent acceptance in its current form. The related work section lacks depth, particularly regarding the role of task diversity in supervised ICL, and the measure of diversity used in this work is not adequately explained. Furthermore, the paper has issues with clarity and presentation, specifically concerning the main figure, the justification for the new ICL benchmark, inconsistencies in the reward function definition, and the readability of these plots. These issues, coupled with the incomplete coverage of existing literature, undermine the overall contribution and novelty of the work. Therefore, we must reject this paper. However, we encourage the authors to address these shortcomings and resubmit a revised version that significantly improves upon the current submission.

**Additional Comments On Reviewer Discussion:**

some questions are addressed by authors, but paper will be benefitted by another revision.

---

### Decision · Program_Chairs · 2025-01-22

Reject